# Optical Coherence Elastography as a Tool for Studying Deformations in Biomaterials: Spatially-Resolved Osmotic Strain Dynamics in Cartilaginous Samples

**DOI:** 10.3390/ma15030904

**Published:** 2022-01-25

**Authors:** Yulia Alexandrovskaya, Olga Baum, Alexander Sovetsky, Alexander Matveyev, Lev Matveev, Emil Sobol, Vladimir Zaitsev

**Affiliations:** 1Institute of Photon Technologies, Federal Scientific Research Center “Crystallography and Photonics”, Russian Academy of Sciences, 2 Pionerskaya Street, Troitsk, 108840 Moscow, Russia; baumolga@gmail.com; 2Institute of Applied Physics of the Russian Academy of Sciences, 46 Uljanova Street, 603950 Nizhny Novgorod, Russia; alex.sovetsky@mail.ru (A.S.); matveyev@ipfran.ru (A.M.); lev@ipfran.ru (L.M.); vyuzai@mail.ru (V.Z.); 3UCI Health Beckman Laser Institute & Medical Clinic, 1002 Health Sciences Rd., Irvine, CA 92612, USA; en.sobol@gmail.com

**Keywords:** optical coherence elastography, optical clearing, cartilage, osmotic strain

## Abstract

This paper presents a recently developed variant of phase-resolved Optical Coherence Elastography (OCE) enabling non-contact visualization of transient local strains of various origins in biological tissues and other materials. In this work, we demonstrate the possibilities of this new technique for studying dynamics of osmotically-induced strains in cartilaginous tissue impregnated with optical clearing agents (OCA). For poroelastic water-containing biological tissues, application of non-isotonic OCAs, various contrast additives, as well as drug solutions administration, may excite transient spatially-inhomogeneous strain fields of high magnitude in the tissue bulk, initiating mechanical and structural alterations. The range of the strain reliably observed by OCE varied from ±10^−3^ to ±0.4 for diluted and pure glycerol, correspondingly. The OCE-technique used made it possible to reveal previously inaccessible details of the complex spatio-temporal evolution of alternating-sign osmotic strains at the initial stages of agent diffusion. Qualitatively different effects produced by particular hydrophilic OCAs, such as glycerol and iohexol, are discussed, as well as concentration-dependent differences. Overall, the work demonstrates the unique abilities of the new OCE-modality in providing a deeper insight in real-time kinetics of osmotically-induced strains relevant to a broad range of biomedical applications.

## 1. Introduction

In the last decades, elastographic imaging technologies that emerged in 1990s have become widely used in various biomedical applications. Such elastographic modalities are based on some methods enabling structural imaging; presently, these are first of all medical ultrasound (US) [1] and magnetic resonance imaging (MRI) [2]. The main medical application of elastographic techniques is characterization of tissue stiffness (i.e., shear or Young’s moduli), for which the basic US or MRI-based imaging is supplemented by some auxiliary mechanical action (either quasi-static or transient) applied to the studied tissues/organs. The transient-type (shear-wave-based) elastography is implemented in several US platforms, enabling ultrasonic visualization of the propagation of auxiliary shear waves in the tissue. In these methods visualization of genuine local strains is not required at all. For elastographic techniques based on quasi-static auxiliary loading, an essential stage of elastographic characterization involves mapping of local slowly varying strains in the studied region. This approach is often called “strain elastography”, so that methods of strain visualization are often called “elastography” even if strains are not necessarily produced by mechanical loading. Visualization of strain requires a series of structural scans of deformed materials/tissues to be obtained. The displacements of particles in the compared images obtained are then recalculated into genuine strains by finding local gradients of the interframe displacements. Due to inevitable measurement errors at both stages (initial reconstruction of displacements and subsequent finding of strains) the resolution of the resultant strain maps is at least several times (and even an order of magnitude) lower than the initial resolution of structural images. Correspondingly, for the conventional US-based and MRI-based elastographic imaging, the resolution is in the order of several millimeters and even lower.

In this context, another visualization technique, Optical Coherence Tomography (OCT), opens up new possibilities, occupying an intermediate niche between high-resolution optical microscopy and medical US in terms of resolution and size of the imaged area. OCT scans represent the distribution of optical-backscattering strength; in their appearance, OCT-scans are rather similar to US-scans, but enable much higher, micrometer-scale resolution. In view of this similarity, by analogy with elastographic approaches that were proposed in the beginning of the 1990s in medical US, utilization of OCT for studying microscopic deformations of biological tissues and assessing their biomechanical properties was proposed ~two decades ago in the seminal paper by J. Schmitt [3]. However, the practical realization of these ideas appeared to be rather challenging, so that breakthroughs in these directions were demonstrated only in the last 5–6 years. Quite often the above-mentioned applications of OCT discussed in [3] are termed optical coherence elastography (OCE), although actually this term is used to denote quite different particular extensions of OCT. Indeed, there are significant differences between visualization of displacements of scatterers and mapping of genuine local strains. There are also significant differences among different approaches to OCT application for assessment of elastic moduli of biological tissues. In comparison with US-based and MRI-based elastographic methods OCT-based elastographic techniques opened a broad range of previously unavailable possibilities, due to the intrinsic much higher resolution of OCE. Since OCT in recent decades has become a standard imaging technique in ophthalmology [4], it is not surprising that OCT-based elastographic strain imaging has also been tested in various ophthalmic applications [5,6,7,8,9,10], and presently is actively studied for oncology-related applications as well [11,12,13].

Some OCE-related works, starting from the pioneering paper by J. Schmitt, by analogy with US-based strain elastography were oriented on utilization of OCT for mapping displacements of scatterers using correlational techniques in various forms, e.g., [14,15]. It was supposed that the initially reconstructed spatial distribution of displacements could then be used for determining local strains by estimating the spatial gradients of the displacements [16,17,18,19]; in other variants strain was mapped in a non-quantitative manner directly using the strain-induced decorrelation [20,21]. An even larger group of works in OCT-based studies of local strains has been stimulated by paper [22] in which utilization of the phase-resolved approach to estimating local strains was proposed. In the present study, we use an advanced realization of phase-resolved OCT-based strain mapping. The method used is termed “vector” because it operates with the OCT-signals represented as vectors in the complex-valued plane. The phased-resolved realization of mapping of local strains and the features/advantages intrinsic to its vector form are explained in more detail in the “Materials and Methods” section. The main point is that OCT-based mapping of local strains enables much higher spatial resolution than that of conventional medical elastography methods (US-based and MRI-based). Thus, OCE opens previously unavailable prospects to quantitatively visualize in real time the osmotic-strain dynamics accompanying penetration of diffusion fronts with a spatial resolution of several tens of micrometers with total observation time of hundreds of seconds and, if necessarily, hours. The characteristic sizes of the imaged area in OCE are the same as in structural OCT imaging, in the order of several millimeters laterally and ~millimeter in the tissue bulk.

Since OCT-based strain imaging may be used to visualize more than just mechanically produced strains, it has been tested to visualize some other types of deformation processes, e.g., tissue drying, heating, etc. [23]. An important class of processes accompanied by mechanical deformations is related to various osmotic phenomena. In particular, such deformations of osmotic origin develop in biological tissues subjected to application of non-isotonic solutions. Among substances demonstrating osmotic activity, much attention is paid to the so-called optical clearing agents (OCAs). Impregnation of biological tissues with such agents is widely used (both in vitro and vivo) to reduce optical scattering in the tissue and thus to increase the penetration depth for optical waves used in various method of optical diagnostics (optical microscopy, OCT, generation of harmonics, etc.) [24]. Substances used as clearing agents may be either isotonic or non-isotonic in comparison with the natural interstitial liquids in the tissue. The same chemical agents (such as glycerol), depending on the concentration, may demonstrate strongly different osmotic properties. Spatially-resolved quantitative characterization of osmotically-induced deformations is a challenging problem, in the context of which the application of the recently developed OCT-based methods of strain mapping opens rich, previously unavailable possibilities.

In this paper we report the first systematic application of OCT-enabled strain mapping for studying osmotic deformations in cartilaginous samples for various types and concentrations of OCAs.

## 2. Materials and Methods

### 2.1. Phase-Resolved Strain Imaging Technique

Since the direct transfer of correlation-based approaches from medical US strain elastography to OCE appeared to be not very efficient, paper [22] proposed an alternative, phase-resolved approach to the realization of OCT-based strain mapping. The approach is based on the well-known relationship between the axial interframe displacement U of scatterers and the resultant variation $\Phi =\phi_{2}-\phi_{1}$ in the OCT-signal phase:(1)$$U=\frac{\lambda_{0}\Phi }{4\pi n}$$

In this equation, $\lambda_{0}$ is the central wavelength of the illuminating OCT signal in vacuum, and n is the refractive index of the examined tissue.

The OCT signal characterized by amplitude A and phase ϕ can be represented as the complex-valued quantity a:(2)a=Aexp(iϕ)=Acos(ϕ)+iAsin(ϕ)

The compared reference and deformed pixelated OCT scans can be represented as a complex-valued matrices $a_{1}(m,j)=A_{1}(m,j)\exp[i\phi_{1}(m,j)]$ and $a_{2}(m,j)=A_{2}(m,j)\exp[i\phi_{2}(m,j)]$, respectively. The interframe phase variation $\Phi (m,j)=\phi_{2}(m,j)-\phi_{1}(m,j)$ can then be found as:(3)$$\Phi (m,j)=\arg\{a_{2}\cdot a_{1}^{*}\}$$

Here, the asterisk * denotes phase conjugation; the matrices are multiplied in the element-by-element sense. The interframe phase variations Φ(m,j) found in this way correspond to the local axial displacements $U(m,j)=\lambda_{0}\Phi (m,j)/(4\pi n)$.

An important limitation of this approach is that the unambiguous relationship between phase variation and scatterer displacement takes place only for sufficiently small displacements, below ½ of the wavelength $\lambda_{0}/n$ in the tissue. For larger displacements, phase wrapping occurs, so that the displacement based on the observed phase variations yield can be reconstructed only with an uncertainly proportional to an integer number of $\lambda_{0}/2n$. To exclude this uncertainty, it is necessary to perform procedures of phase unwrapping. To reconstruct axial strains ∂U/∂z based on the displacements estimated in this way, in [20] it was proposed to estimate the axial gradients ∂Φ/∂z∝∂U/∂z of phase variations using the least squares method for determining the slope of the dependence Φ(z).

Later, an alternative “vector method” of estimating the phase-variation gradients was proposed in [25,26,27], where this approach is described in detail. The name “vector method” reflects the fact that it treats the OCT signal with amplitude and phase as vectors in the complex-valued plane without the necessity of explicitly single out the phase till the very last step of the transformations. Besides significantly better computational efficiency, this method enables a number of other advantages: it is intrinsically very robust with respect to various measurement errors (the method is intrinsically able to suppress the strongest phase errors ~π rad. and gives convenient possibilities for amplitude weighting to reduce the especially noisy contributions of small-amplitude pixels). This method also obviates the necessity of phase unwrapping even if the displacements are essentially on the supra-wavelength scale.

In combination with the application of pre-calibrated layers of translucent silicone, the vector method of strain mapping enables convenient possibilities for realization of compression OCE (C-OCE) to realize spatially resolved mapping of the Young’s modulus of the tissues examined [28,29,30]. Discussion of numerous examples of biomedical applications of the Young’s modulus mapping based on C-OCE can be found in the recent review [23]. However, the possibility of spatially-resolved mapping of strains based on the analysis of interframe phase variations, even independently of elasticity mapping, opens very interesting prospects for studying a broad range of processes. In particular, non-trivial features of thermally-produced strains in collagenous tissues were found in [31]. The OCE-based strain mapping was also specially adapted for detection and characterization of slow deformations related to relaxational phenomena [32,33], this adaptation being important for longitudinal visualization of slow-rate stages of OCAs penetration in the results presented below.

### 2.2. Preparation of Cartilaginous Samples and Strain-Measurement Conditions

As a representative example of poroelastic collagenous tissue, the transverse sections of porcine costal cartilage were used. Transverse cartilage sections contain approximately parallel oriented collagen fibers and their bundles [34] which are symmetrically distributed around the cartilage length axis (Z), so that the tissue can be considered more or less structurally isotropic in the Z-plane. The porcine costal cartilages of 5th–8th ribs were taken from a local butcher immediately after slaughter and stored frozen at −15 °C. Thawing was performed stepwise: first, at 4 °C for at least 8 h, then at room temperature for 1 h. Prior to the measurements, all samples were equilibrated in saline solution containing 0.9% NaCl (~300 mOsm). The cross-sectional cylindrical cuts of cartilages with d ≈ 10 mm and thickness 2.0 mm were prepared using a scalpel and metal punch tool. The axes of cylinders were oriented along the *Z*-axis of OCT scans.

The osmotically-induced strain dynamics were studied using a custom-made spectral-domain OCT setup operating at a central wavelength of 1300 nm (~90 nm spectral width), 20 kHz rate of obtaining spectral fringes, and 20 Hz rate of acquiring B-scans covering 4 mm laterally and 2 mm in depth (in air). Initially, a series of complex-valued OCT-scans of the tissue experiencing deformations was acquired and, if necessary, the inter-B-frame interval could be increased to improve conditions for studying sufficiently slow deformations, as discussed in [32,33]. The series of OCT-scans thus acquired enabled depth- and laterally-resolved 2D maps of interframe- and cumulative strains to be obtained using the estimation of local axial gradients of interframe phase variations [23,31]. The samples were placed in an equilibrating bath (Figure 1) and at the start of OCT recording a ~1 mm layer of solution was poured onto the cartilage surface.

Glycerol-water solutions of various concentrations, and the commercially available contrast agent Omnipaque-300 (300 mg/mL iodine, 39.2% mass iohexol solution) GE Healthcare (Cork, Ireland), were used as osmotic agents. To study the strain associated with elution of an osmotic agent, cartilage samples were equilibrated in 50% glycerol for 48 h and then the measurement was performed by pouring saline solution into the experimental bath. The obtained data were analyzed using custom-made software for strain visualization based on the vector algorithm described in [25,26,32] for mapping strains. The processing-window, over which the strain was estimated and visualized in elastographic B-scans, had the sizes 100 × 100 μm in the axial and lateral directions. Consequently, the spatial resolution in the elastographic images corresponded to ~1/2 of the processing window, i.e., ~50 μm in both directions. Since the strain distribution in the studied samples had approximately plane-layered structure oriented laterally, for plotting depth dependences of strain, lateral averaging could be made over a larger size to suppress small-scale fluctuations in the lateral direction. Correspondingly, the depth dependences extracted from the initially plotted strain maps was made using the averaging window with Z-X dimensions of ~80 × 800 µm. The interframe time interval was 1 s. The results are given as maps of cumulative strain representing the results of summation of interframe strains as a function of the observation time.

All measurements were repeated at least 3–4 times, the results reflect the representative findings. The following features of the measurements can be pointed out: (1) strain fields were obtained for central cross sections (with the diameter 8–10 mm) of costal cartilage, whereas the sites near the tissue borders with non-isotropic structure along the *Z*-axis were not studied; (2) the clearing effect of the substances used, especially that of highly concentrated glycerol solutions, additionally decreased the level of the OCT signal received from the deeper layers of the sample, therefore the depth analysis was limited to 600–900 µm below the sample surface; (3) for low-concentrated glycerol solutions and glycerol elution experiment with the osmotically-induced strain not exceeding 10^−4^, the averaging window size was increased to ~160 × 800 µm when plotting the depth profiles to decrease the influence of noise; (4) the osmotic-strain dynamics in the present study correspond to the non-equilibrium regime in which the observed strain is accompanied with the mass fluxes outside and inside the sample, since the time of continuous observation is 10–15 min from the moment of the sample comes into contact with the solution.

## 3. Results

### 3.1. Types of Osmotic Effects

Various osmotic effects studied by OCE are presented in Figure 2. For sample groups shown in (Figure 2a–c), when the solute concentration is not strongly hypertonic, a thin region (~60–80 µm) of subsurface shrinkage can be seen as an intense blue-color layer just below the sample surface in the cumulative strain images (Figure 2(a-2,b-2,c-2)). Such an effect can be attributed to the diffusion of near-surface tissue macromolecules into solution from the cut surface. The mean amplitude of this subsurface shrinkage was determined to be 0.05 ± 0.02 for isotonic (Figure 2a) and hypotonic (Figure 2b) saline and 0.06 ± 0.02 for 7% weakly hypertonic glycerol solution (Figure 2b).

For distilled water (data not shown), the subsurface shrinkage value is slightly lower: 0.04 ± 0.01, which indicated the possible participation of solute molecules (ions) in desorption mechanism of macromolecule residues from the cut surface.

Further estimations of osmotic effects were made for the deeper layers in the tissue bulk located below the above-mentioned thin layer of the subsurface shrinkage in Figure 2(a-2–c-2). In Figure 2(d-2), for which the glycerol concentration exceeds ~30%, the amplitude of the osmotic-induced subsurface dilatation (shown by the red color) reaches 0.1–0.15 and completely masks the much weaker subsurface shrinkage effect visible in Figure 2(a-2–c-2).

The 0.9% aqueous solution of NaCl (saline) applied to the samples is used as an isotonic solution in medicine to maintain hydration of biological tissues with minimal alterations of their physiologic osmolarity. Here saline is used as a reference substance to visualize strain dynamics in cartilage in nearly-isotonic conditions (Figure 2a). According to the cumulative strain image (Figure 2(a-2)), the strain depth profile in Figure 2(a-3) and 2D “waterfall” kinetics shown in Figure 2(a-4)) the maximal accumulated strain for 300 s of equilibration of cartilage in saline does not exceed 3∙10^−4^ and for depths smaller ~300 µm this small strain is negative, whereas for deeper layers, it changes its sign from negative to positive (keeping the small magnitude).

This result is only slightly different from that for distilled water (data not shown), for which the small magnitude of ~2∙10^−4^ of maximal accumulated strain was observed with the sign changing from negative to positive at a depth of about 400 µm. The effect of weak axial shrinkage of subsurface layers within 300–400 µm is probably caused by the imbalance of overall polymer concentration in cartilage and solution: when a sample with a certain polymer concentration is put into polymer free solution the diffusion of polymers should occur from cartilage to solution to equalize their concentrations in the two phases [35,36]. The diffusion of macromolecules from the subsurface layer of cartilage is easier than that from the bulk where they are held by intermolecular bonds. Therefore, only a thin subsurface layer of intense shrinkage in diluted solutions is observed in the immediate vicinity of the surface where the tissue structure is damaged by the surgical cut (see the narrow blue near-surface layer in Figure 2(a-2,b-2,c-2)). On the contrary, for somewhat deeper layers, the shrinkage effect is comparatively weak: for the times of observation of 300–600 s, small molecules can be washed out by the diffusion of the interstitial fluid.

The comparison between the effect of saline and distilled water indicates that the balance of the ionic component of saline does not strongly affect the observed axial strain for equilibration times of several minutes (up to 10–15 min) and does not make a noticeable contribution to the rapid and strong deformation effects caused by application of the highly concentrated solutions used in this study (an example of which is shown in column (d) of Figure 2).

In this context, Figure 2c,d shows examples of the effect of glycerol solution on the strain dynamics in cartilage for two essentially different glycerol concentrations. Glycerol is known as one of the most common clearing agents for various applications [24,37]. Its clearing mechanism is well understood for the skin and other collagenous tissues [38,39,40]. For high concentrations, it implies strong dehydration, shrinkage, and more regular packaging of the tissue collagen [40]. For clearing of costal cartilage, glycerol was used in [41], showing a noticeable tissue shrinkage effect for 1.5 h immersion time.

In the present study, the glycerol-induced strain field in costal cartilage is studied for the first few minutes of immersion. In terms of non-equilibrium strain evolution immediately after the application of clearing solutions, the two principal effects are observed depending on glycerol concentration. For diluted glycerol solutions (see Figure 2(c1–c4)), the dilatation effect prevails throughout the bulk of the tissue. On the contrary, for high glycerol concentrations, the alternating-sign strain profile occurs (Figure 2(d1–d4)). For 7% (*w*/*w*) glycerol (Figure 2(c1–c4)) the maximal observed strain reaches +2.5∙10^−3^ for 600 s of observation, which is an order of magnitude higher than that obtained for saline (Figure 2(a1–a4)) or for distilled water. The application of 35% (*w*/*w*) glycerol results in a dilatational strain with the magnitude ~0.2 at smaller depths and then for deeper layers, shrinkage with strain magnitude ~0.1 occurs (see Figure 2(d-2,d-3)). These effects are two orders of magnitude stronger than for 7% (*w*/*w*) glycerol ((see Figure 2(c-2,c-3))) and three orders of magnitude stronger than for saline (Figure 2(a-2,a-3)).

Simultaneous strong dehydration-induced shrinkage in the bulk and subsurface dilatation of the tissue caused by the hyperosmotic effect of high-concentration glycerol can be seen in Figure 2d. According to the cumulative strain image (Figure 2(d-2)) and depth strain profile (Figure 2(d-3)), the dilatation (positive strain) is observed within ~320 µm subsurface layer and is followed by a deeper layer of strong shrinkage (negative strain). The more detailed description of glycerol concentration effects on alternated-sign strain evolution is given in the next section.

The weak hypotonic regime was modeled by application of saline to the cartilage samples preliminary saturated with 50% (*w*/*w*) glycerol solution (Figure 2b). Here, the thin layer of subsurface intense shrinkage is observed similarly to a near-isotonic case (compare Figure 2(a-2,b-2)) and the strain amplitude is 0.05 ± 0.02, which is similar to that for cartilage in the native state immersed in saline (Figure 2a). The main flux process in hypotonic conditions shown in Figure 2b should be the equalizing of glycerol concentration in interstitial fluid and solution. The diffusion of glycerol molecules is slower than that of water. Additionally, the tissue permeability for glycerol was estimated to be three orders of magnitude lower than for water [38]. Thus, the total amplitude of negative strain for 600 s is not high, with an extreme value of about −1.6·10^−3^ (Figure 2(b-2)). However, such hypotonic-induced shrinkage is 5 times more intense than the shrinkage effect observed for saline equilibration.

### 3.2. Strain Dependence on Glycerol Concentration

Various characteristic scenarios of strain evolution in cartilage depending on glycerol concentrations are shown in Figure 3 in more detail. Four glycerol concentrations are considered: 7, 25, 35, and 100% (*w*/*w*). The cumulative strain images for 600 s of observation are given in Figure 3a–d. Kinetics of strain minima and maxima for the solutions are given in Figure 3e,f, correspondingly. Dynamics of the strain extreme values show a particularly strong dependence on concentration (Figure 3e,f). For glycerol concentrations from 7 to 35%, the maximum of axial dilatation increases by two orders of magnitude from ~0.002 to 0.2 (Figure 3f) and the magnitude of axial shrinkage (negative maximal strain) due to dehydration effect also strongly varies, from −0.006 for 25% glycerol to −0.14 for 35% (Figure 3e). The pure 100% glycerol solution especially demonstrates a strong increase in the dehydration effect: the negative strain for 100% glycerol is −0.36, which is ~2.5 times more intense than that for 35% solution (Figure 3e). The difference in the positive strain intensity increases somewhat smaller, only two times from 0.20 for 35% concentration to ~0.4 for 100% glycerol concentration (Figure 3f).

Depth dependences for strains corresponding to various glycerol concentrations are shown in Figure 4a–d. One can see that when the dehydration effect is comparatively weak, as for 7% and 25% glycerol concentrations (Figure 4a,c), the axial expansion is fairly uniformly spread throughout the observable depth to ~700–800 µm. The dehydration-related shrinkage is observed in subsurface layer not exceeding 100 µm, though for 25% glycerol it appears to be significantly stronger than for 7% (Figure 4a,c). For 35% glycerol solution, expansion (positive strain) is not so uniformly distributed as for smaller concentration; on the contrary, it is localized in a thinner subsurface layer down to ~450 µm, whereas the dehydration maximum is clearly observed at deeper layers and does not manifest itself at small depths just below the surface (Figure 4b). For 100% glycerol, the subsurface minimum again becomes well manifested. The limited depth range of positive strains is also well visible, such that the strain profile crosses the neutral line at two depths: at ~100 µm from negative to positive strain and at ~400 µm from positive to negative strain.

Figure 5 shows the time evolutions of the depth positions of strain minima and maxima for different glycerol concentrations. For high concentrations of glycerol that cause strong dehydration, the positive strain maxima are located within a fairly narrow depth interval ~100–200 µm below the surface, slowly migrating to the deeper layers (Figure 5a,b). In comparison to the strain maxima, the kinetics of strain minima is more intense: their depth positions move from ~200 to ~700 µm for 100% glycerol (Figure 5a) and from 200 to 800 µm for 35% glycerol (Figure 5b). In both cases, the neutral line position in the course of this evolution remains approximately in the middle between the positions of the strain minimum and maximum. For low glycerol concentrations and absence of strong dehydration, the strain maxima remain steadily located in the vicinity of ~200 µm depth (Figure 5c,d). At ~400 s of observation the strain-maximum position gets shifted towards ~450 µm for 7% glycerol (see Figure 5d). However, it is clear from Figure 4a that for 7% solution, the distribution of the positive strain is rather broad and flat, so that the shift visible in Figure 5d corresponds to only a slight modification of the shape of the broad maximum shown in Figure 4a. The negative strain for low glycerol concentrations from the start of observation is shifted towards far deeper layers, closer to the detection limit (Figure 5c,d), so that for 7% glycerol, the position of the too deep minimum cannot be determined correctly. For 25% solution, the strain minimum is better detectable and is more or less steadily located between 700 and 900 µm (Figure 5c). The neutral line for 25% is also shifted to the deeper layers towards the strain minima, which indicates the prevalence of axial extension throughout the observable thickness (Figure 5c).

### 3.3. Strain Dependence on the Type of Osmotic Agent

Glycerol is a strong hyperosmotic agent which at high concentrations may affect not only the tissue cell viability, but even the integrity of collagenous structure. The molecular weight of glycerol is relatively small, at M_w_ = 92 g/mol, meanwhile it has three OH groups with the ability to form hydrogen bonds and effectively bond water molecules. Small concentrations of glycerol are ubiquitously used as additives in the pharmaceutical and cosmetics industries. However, the clearing ability of low concentrations of glycerol is usually not sufficient [38], therefore it is important to consider agents with higher concentrations and moderately stronger osmotic effects.

In Figure 6, the effect of ~40% aqueous iohexol (M_w_ = 821 g/mol) solution (Omnipaque) on cartilage is shown. The alternated-sign strain visible in Figure 6b is qualitatively similar to the case of glycerol solutions of 35% and 100% concentrations. The absolute value of strain in Figure 6c does not exceed 0.15. The dehydration effect of Omnipaque is substantially less pronounced, so that the magnitude of strain minimum at 600 s amounts to −0.06 (Figure 3e and Figure 6c) and is located comparatively close to the surface: Figure 5e shows that both extreme values lay within ~300 µm under the surface. The zone of dehydration with negative osmotic strain spreads towards ~500 µm depth (Figure 4e).

## 4. Discussion

In the present work, the ability of the recently developed modality of phase-resolved OCT to map osmotically-induced deformations in biological tissue has been demonstrated. The method allows one in a non-contact manner to obtain strain distributions resolved in the tissue depth and laterally during diffusion of optical clearing agents. Alterations of tissue mechanics under the action of various OCAs have been studied for decades [24]. However, the majority of works were focused on the differences between equilibrated states: native tissue and tissue saturated with corresponding OCA to the equilibrium state [42,43,44,45]. Only a few recent works analyzed what happens during the clearing process, when the OCA diffusion is accompanied by evolution of mechanical moduli [46]. In [46], the effect of 20% glucose on hyaline-type cartilaginous tissue was investigated and it was found that, for the first 10–15 min of diffusion, the tissue stiffness rapidly decreased, while for the rest of the 110 min it gradually restored its initial value. This finding demonstrates further evidence of the importance of the processes occurring in the first few minutes of immersion.

As an instructive example of a study related to the effect of non-isotonic solutions on a collagenous tissue, one can mention paper [47], where phase-sensitive OCT was used to observe the de-swelling/swelling osmotically-induced phenomena occurring in eye cornea tissue following storage for corneal transplantation. In this study the osmotically active hypertonic substance was dextran solution rather than the OCA solutions used in the present study. However, for comparison with the present study, it should be pointed out that in [47] quite large strains of the tissue were also observed, because the studied corneal samples could change their thickness ~two times during the osmotic de-swelling/swelling. These data agree with the cumulative strain values up to several tens per cent observed in the present study. Another generically similar observation in [47] is that during the observation interval the tissue may behave non-monotonically, for example demonstrating initially decrease in the thickness and then increase. Similarly to the present study, the strongest deformations were observed at the first stages of the osmotically-induced filtration of the liquids. Finally, alternating-sign deformations were observed both depending on the elapsed time and observation depth within the cornea thickness.

In the present work it is also shown that deformations of the highest magnitude are observed within the first few minutes of glycerol diffusion, and proper understanding of this effect is important from the viewpoint of the overall efficiency and biological safety of the clearing procedures, and also because in some cases it may affect the tissue integrity.

In this context, one should distinguish the average (“global”) deformations of the macroscopic tissue volume subjected to optical clearing and local spatially non-uniform strain variations caused by high gradients of OCA concentration. The first process involves the alteration of the macroscopic parameters of the tissue, such as thickness and length, or bulk moduli [41,42,43,44,45,48]. For instance, cartilaginous tissue demonstrates extensive shrinkage after ~1.5 h application of clearing solutions of glycerol and glucose [41] which is already available for visual assessment and can be directly measured by conventional optical microscopy ruler with sufficient accuracy. The knowledge of such a durable cumulative effect of OCA on the tissue structure is of the upmost importance for estimation of its properties at the final stage of clearing, which is close to the equilibrium state in terms of diffusion. However, in the first minutes of OCA diffusion, the distributions of water and agent concentrations are strongly non-uniform [49,50]. Additionally, the so-called diffusion fronts occur, which may be accompanied by intense straining and dramatically alter tissue mechanics, though the existence of these transient effects is comparatively short-term [47]. These non-equilibrium effects are insufficiently studied up to the date due to the complexity of theoretical analysis and the lack of physical methods enabling their direct observation. As shown in the present work for cartilage, and in [47] for the cornea, such osmotically-induced non-uniformities of strains and displacements can emerge on a scale of tens of micrometers. In fact, the effects of the non-equilibrium strain fields on cell viability, matrix integrity and sustainability of different tissues have never been directly observed before. It may reasonably be expected that the duration of a non-equilibrium state is dependent on the size of a particular agent molecule. This expectation is supported by the present study, in particular Figure 3f, which demonstrated that the osmotically-induced positive strain fields in the case of small glycerol molecules become nearly saturated within 200–350 s intervals, whereas for Omnipaque, strain still continues to gain intensity (Figure 3f). For dextran, a high molecular weight compound studied in [47], it takes tens of hours to proceed through maximum strain intensity caused by dehydration and dextran integration. It was shown previously on isolated chondrocytes and 3D cartilaginous scaffolds that alterations of micromechanical properties of the matrix tissue surrounding cells may influence to a great extent to the cell signaling and overall tissue homeostasis [51,52,53]. The recently found mechanotransduction pathways in cartilage have attracted increasing attention in recent years [54]. Moreover, osmotic stress may influence the metabolic activity of cells [55], in extreme cases leading to cell death [56]. In this regard, the method of direct and non-destructive monitoring of transient mechanical strains demonstrated in the present work may greatly contribute to the evaluation of the particular correlations between the osmotically-induced strain duration and intensity and biochemical response of the tissue. Moreover, it is noteworthy that the method is not limited to cartilaginous tissues, and the scope of its applications can be significantly expanded to other tissues and organs where superficial scanning is accessible.

Therefore, time-resolved monitoring of the effects of a particular OCA concentration on strain evolution within the tissue in the first minutes of diffusion can provide additional important data on the clearing mechanisms and reveal possible adverse effects of OCA administration.

Note also that according to Equation (1) for the used phase-sensitive OCE modality, the axial displacement is directly proportional to interframe phase variation. In principle, this phase variation depends not only on mechanical displacements of scatterers, but potentially may be affected by variations in the refractive index. Thus, the strain evolution maps shown in Figure 2, Figure 3 and Figure 6 rigorously speaking should reflect the interplay of these mechanical and optical effects. When refractive index is constant, i.e., when the influences of variations in the interstitial fluid concentration and volumetric content are negligible, the observed phase variations depend only on interframe displacements of scatterers, and their axial derivative represents the actual axial strain. However, in a non-equilibrium regime when gradients of OCA concentration are fairly large, certain variations in the refractive index may occur. It is the mismatch of refractive indices between the interstitial fluid and solid tissue components, first of all, collagen fibrils, which is mainly responsible for the tissue opacity in visible and near infrared spectral range [57]. The effect of optical clearing by application of hypertonic OCAs, such as glycerol, is mostly related to reduction in the refractive indices mismatch throughout the tissue, thus providing deeper penetration of optical radiation [24,57].

In this context it is important to understand to what degree the refractive index variations during the first steps of the clearing process may affect the apparent strain evolution in the phase-sensitive OCE images. The most dramatic drop of *n* may occur on the border of fluid and dried solid matter (protein): from 1.330 to 1.598, respectively, according to the data presented in Table 2 of [57]. This accounts for approximately 20% of the total *n* variation. In fact, the experimental conditions of the present study exclude the potential total dehydration of the tissue, for it is always saturated with liquids of different contents: saline, glycerol-water solutions, or Omnipaque. Furthermore, glycerol-water concentrations less than 30% *w*/*w* do not cause intense dehydration of collagen fibers [38]. The refraction diagram of water-glycerol system corresponds to the gradual increase in *n* with increasing glycerol concentration (0–99% *w*/*w*) from 1.33 to 1.47 [58]. Thus, the most dramatic variation in *n* that may be expected near the interface with pure glycerol amounts to ~10%. However, this may be true only for extremely high glycerol concentration about 100% (*w*/*w*) (Figure 3d), whereas in other cases the mismatch should be significantly lower. For example, the refractive index of 25% (*w*/*w*) glycerol is only slightly different from that of water, 1.36 versus 1.33, respectively [58]; for 35% glycerol the ratio is 1.38/1.33. From 10 to 100% (*w*/*w*) glycerol concentration *n* value grows by ~10%, whereas the observed positive-signed strain maximum in cartilage increases by a factor of 160 when glycerol concentration is increased from 7 to 100% (*w*/*w*) (Figure 3f). For strain minima, the negatively signed strain magnitude grows by a factor of 65 (Figure 3e) versus 8% growth of corresponding *n* values from 25 to 100% (*w*/*w*) glycerol [58]. Therefore, with the change of glycerol concentration, the observed strain magnitude varies more strongly than the accompanying variation in the refractive index *n* caused by possible protein dehydration. Indeed, for the diffusion of pure glycerol presented in Figure 3d, the observed cumulative-strain magnitude for 5 min of immersion exceeds 0.4 for positive-signed strain and −0.3 for negative-signed strain (Figure 3d–f), which is a noticeably greater than even the extreme estimate of variation in *n* and far more intense variation than real alterations of *n* for the reported experimental conditions.

The investigation of the glycerol diffusion coefficient in cartilage gives 10^−9^ m^2^/s for 90% (*v*/*v*) glycerol [59], so that for 5 min the expected effective penetration depth of glycerol is about ~$\sqrt{Dt}$, i.e., ~0.6 mm. This value is comparable to the maximal observable depth in the OCE experiment. Thus, during the interval of observation, glycerol is expected to penetrate almost the whole observable thickness of cartilage. In view of this, the alternating-sign strain evolution at the subsurface level for highly hyperosmotic glycerol solutions (>30% *w*/*w*) most probably is not associated with the solute “integration” into the tissue matrix, whereas the kinetics of tissue dilatation for diluted glycerol (<25% *w*/*w*) may reflect the real process of glycerol diffusion (Figure 3a,c and Figure 4a,c).

Consequently, the alternating-sign nature of strain evolution in cartilage during diffusion of highly hyperosmotic glycerol solutions (>30% *w*/*w*) is most probably connected with complex processes of the interstitial structural transformations including matrix dehydration manifesting as tissue shrinkage and subsurface swelling. Concerning possible reasons of this one can argue that, first, the hindering for glycerol permeation into the tissue matrix due to its dehydration-related shrinkage leads to its anomalous accumulation at the subsurface level and subsequent tissue dilatation. The second reason can be connected with the structure of the cartilaginous matrix itself. The porous matrix of hyaline-type cartilage consists mainly of collagen and proteoglycan aggregates integrated with each other according to the network principle [60]. Proteoglycan molecules possess the negatively charged sulfate and carbonate groups that are fixed and in normal conditions are compensated with the positive ions of interstitial fluid. The violation of matrix electroneutrality due to, for instance, compressive load and outflow of the fluid with compensating ions from the compressed area, gives rise to a series of forces striving to restore the equilibrium, which include streaming and diffusion currents, changing of the macromolecule conformation, and even piezoelectric effects [61]. Considering the phenomena of alternating-sign character of strain in the subsurface cartilage layer during the first minutes of glycerol diffusion observed in the present study, one can speculate that the strong subsurface dehydration of cartilage caused by the application of glycerol may also promote the outflow of positively-signed ions that can no longer compensate for the negatively-signed cartilage proteoglycan groups. Further, the electrostatic repulsion of the “naked” charges of proteoglycans may cause the process similar to that known as cartilage osmotic regulation [60,62,63]. The electrostatic repulsion of the liberated fixed charged groups may provoke the dramatic structural extension shown in (Figure 2c and Figure 3b,d). Further research appears to be needed to reveal the nature of glycerol-induced sign-alternating strain fields in cartilage. In this context the described OCE-based technique opens rich possibilities unavailable to the previously used experimental techniques.

## 5. Conclusions

The application of the new OCE modality to observe non-uniform osmotically-induced strain fields in biological tissues is presented. OCE enables one to monitor strain alterations as small as 10^−4^ and as large as 0.4–0.5 for axial strain value. The duration of uninterrupted observation can be tens of minutes, which allows one to cover the whole range of the most intense strain accumulation during OCA diffusion into biological tissue. The concentration-dependent effects studied for glycerol impregnation into cartilaginous tissue include low intensity subsurface dilatation for concentrations less than 30–35% and an alternating-sign strain field for highly concentrated glycerol. The effect of Omnipaque was found to be similar to that of glycerol but proceeding with substantially slower rate.

## Figures and Tables

**Figure 1 materials-15-00904-f001:**
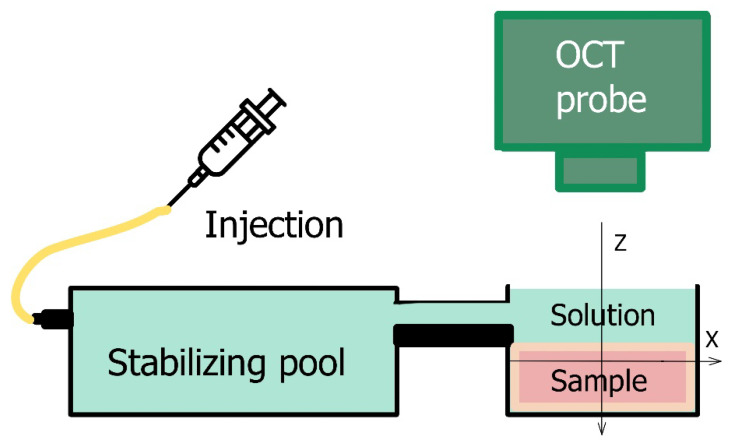
Experimental set up for osmotically-induced strain monitoring.

**Figure 2 materials-15-00904-f002:**
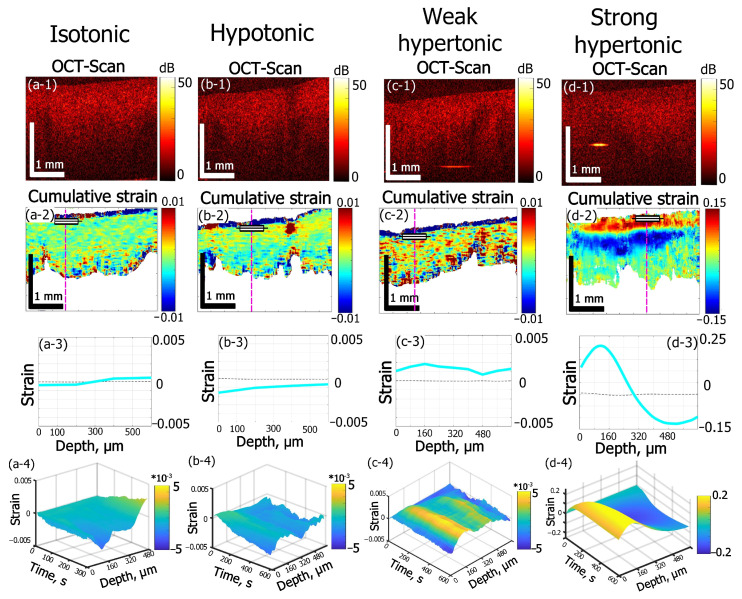
Effects of solutions of different osmolarity on strain kinetics and amplitude inside cartilage samples. The row numbers from 1 to 4 correspond to the following images: (1) structural OCT scans; (2) the corresponding cumulative-strain in-depth maps, (3) strain distribution along the dashed lines in the images in row (2); (4) “waterfall” diagrams of strain dynamics along dashed lines in (2). The columns from (**a**–**d**) correspond to: (**a**) saline solution (0.9% NaCl, 300 mOsm) applied to cartilage samples in the native state, time of observation 300 s; (**b**) saline solution (0.9% NaCl, 300 mOsm) applied to cartilage samples preliminarily saturated with 50% glycerol, time of observation 600 s; (**c**) 7% aqueous weakly hypertonic glycerol solution, time of observation 600 s; (**d**) 35% strongly hypertonic aqueous glycerol solution, time of observation 600 s. Notice that for row 3 showing the depth profiles for the cumulative strain, the depth is counted from the rectangular markings shown in the images in row 2.

**Figure 3 materials-15-00904-f003:**
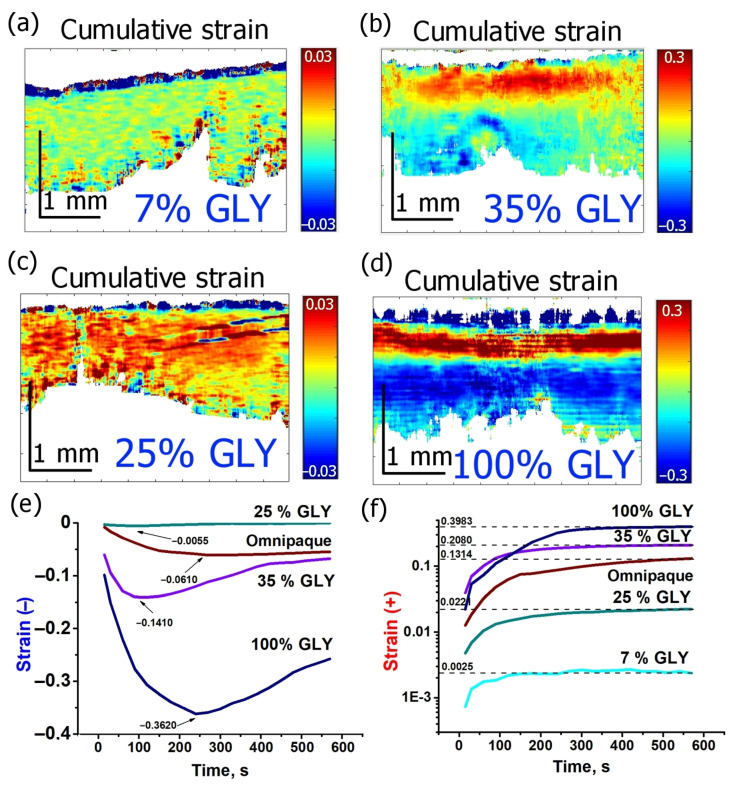
Evolution of strain in cartilage induced by various concentrations of glycerol and Omnipaque: (**a**–**d**)—cumulative strain images for 600 s strain accumulation under glycerol solutions of 7, 35, 25 and 100% (*w*/*w*), correspondingly, (**e**)—kinetics of strain minimum (shrinkage), (**f**)—kinetics of strain maximum (dilatation); In (**f**) the vertical axis is in logarithmic scale.

**Figure 4 materials-15-00904-f004:**
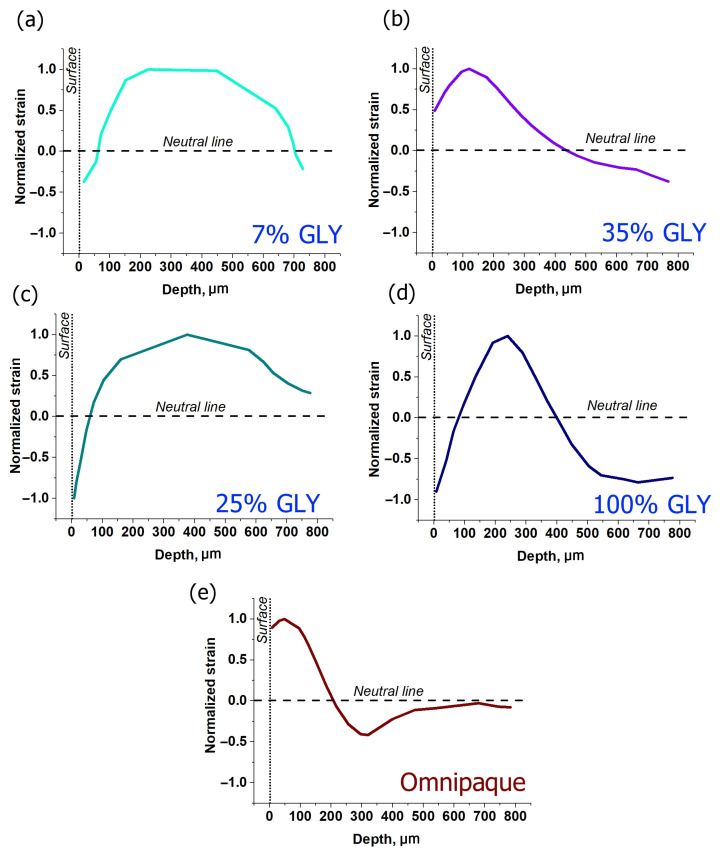
Normalized strain distributions along the cartilage depth for various clearing solutions obtained for 600 s of strain accumulation: (**a**–**d**)—glycerol solutions of 7, 35, 25 and 100%, correspondingly, (**e**)—Omnipaque. Normalizing was carried out in relation to the maximal value of positive strain.

**Figure 5 materials-15-00904-f005:**
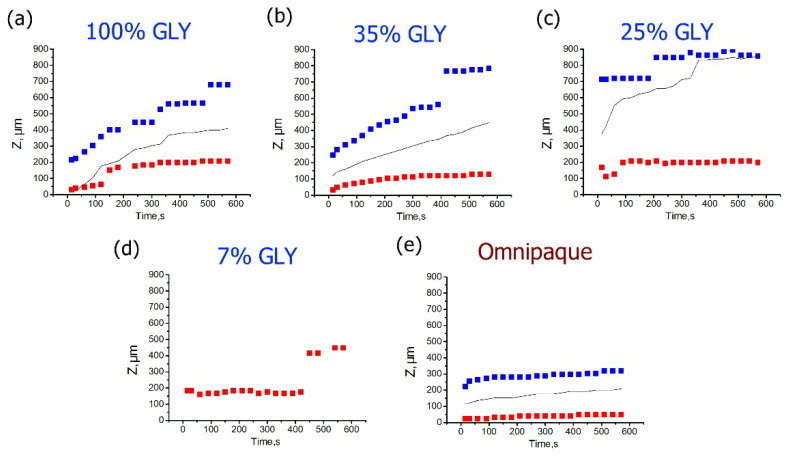
Time dependences for the depth positions of the cumulative strain minima (blue) and maxima (red) corresponding to various glycerol concentrations for 600 s observation time. (**a**–**d**) are for the glycerol solutions of 7%, 35%, 25%, and 100% (*w*/*w*), correspondingly, (**e**) is for Omnipaque. The data for the neutral line of zero strain are shown by solid black curve.

**Figure 6 materials-15-00904-f006:**
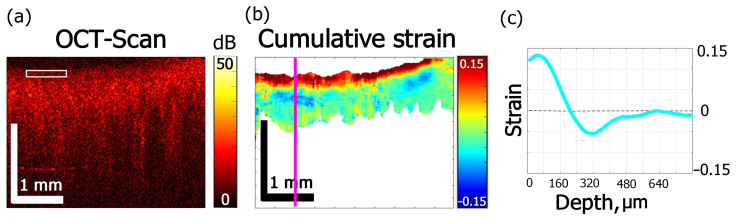
Effects of Omnipaque on strain kinetics and amplitude inside cartilage obtained for 600 s: (**a**) OCT structure image, (**b**) cumulative strain image, (**c**) strain distribution along the line in (**b**).

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
