# Peer review of "Optical Coherence Elastography as a Tool for Studying Deformations in Biomaterials: Spatially-Resolved Osmotic Strain Dynamics in Cartilaginous Samples"

_materials, 2022, doi:10.3390/ma15030904_

Round 1

Reviewer 1 Report

Paper constitutes a worth noticing work prepared in a scientifically proper style. The topic is worth investigating in viewpoint of the research focused on the development of biomaterials. The only suggestions concern mainly editorial aspect of the publication. All comments are given below.

  • Abstract should be supplemented with some numerical data from performed experiments.
  • From editorial viewpoint, the notations of references such as [14, 15, 16] should be replaced by the notation [14-16]. It applies to the entire paper.
  • Caption of Figure 3. is significantly too long and should be shortened.
  • Quality of Figure 2,3 and 6 needs to be improved; additionally these figures should be enlarged to be better readable.
  • Manuscript needs to be supplemented with separate section entitled “Conclusions” in which the most important highlights of performed works will be briefly indicated.
  • Section References should be improved to be consistent, e.g. references 21, 30-32, 38-39 and 41 contain the whole journal names instead of their abbreviations.

Author Response

Reviewer 1 – General comment

Paper constitutes a worth noticing work prepared in a scientifically proper style. The topic is worth investigating in viewpoint of the research focused on the development of biomaterials. The only suggestions concern mainly editorial aspect of the publication. All comments are given below.

Authors: We greatly thank Referee 1 for the effort and time spent for assessing our manuscript and greatly appreciate the constructive feedback and suggestions which helped us improve the quality of the manuscript. In the revised manuscript all introduced changes are highlighted by blue-color font.

Specific comments

1) Reviewer: Abstract should be supplemented with some numerical data from performed experiments.

Authors: Abstract has been supplemented with the following statement containing numerical results: “The range of the strain reliably observed by OCE ranged from ±10-3 to ±0.4 for diluted and pure glycerol, correspondingly”

2) Reviewer: From editorial viewpoint, the notations of references such as [14, 15, 16] should be replaced by the notation [14-16]. It applies to the entire paper.

Authors: The notations have been edited according to the reviewers comment throughout the manuscript.

3) Reviewer: Caption of Figure 3. is significantly too long and should be shortened.

Authors: Fig.3 caption has been shortened to the following: “Evolution of strain in cartilage induced by various concentrations of glycerol and Omnipaque: (a),(b),(c),(d) – cumulative strain images for 600 s strain accumulation under glycerol solutions of 7, 35, 25 and 100% (w/w), correspondingly, (e) – kinetics of strain minimum (shrinkage), (f) – kinetics of strain maximum (dilatation); In (f) the vertical axis is in logarithmic scale.”

4) Reviewer: Quality of Figure 2,3 and 6 needs to be improved; additionally these figures should be enlarged to be better readable.

Authors: In the revised manuscript the size of subplots in Figures 2,3 and 6 was increased and the resolution improved.  

5) Reviewer: Manuscript needs to be supplemented with separate section entitled “Conclusions” in which the most important highlights of performed works will be briefly indicated.

Authors: Conclusion briefly summarizing the main results has been added (Sec.5): “The application of the new OCE modality to observe non-uniform osmotically-induced strain fields in biological tissues is presented. OCE enables one to monitor the strain alterations as small as 10-4 and as large as 0.4-0.5 for axial strain value. The duration of uninterrupted observation can be tens of minutes which allows one to cover the whole range of the most intense strain accumulation during OCA diffusion into biological tissue. The concentration-dependent effects studied for glycerol impregnation into cartilaginous tissue include the low intense subsurface dilatation for concentrations less than 30-35% and alternating-sign strain field for highly concentrated glycerol. The effect of Omnipaque was found to be similar to that of glycerol, but proceeding with substantially slower rate.”

6) Reviewer: Section References should be improved to be consistent, e.g. references 21, 30-32, 38-39 and 41 contain the whole journal names instead of their abbreviations.

Authors: References have been improved according to the reviewers comment.

Reviewer 2 Report

This paper presents an application of phase-resolved Optical Coherence Elastography (OCE) to visualize transient local strains in biological tissues. Applications of this technique are demonstrated for studying the dynamics of osmotically-induced strains in cartilaginous tissue with optical clearing agents. The technique is able to resolve transient spatially-inhomogeneous strain fields excited by the clearing agents, to reveal details of complex spatio-temporal evolution of alternating-sign osmotic strains during agent diffusion. Differences among different clearing agents are discussed with respect to mechanical mechanisms associated with real-time kinetics.

This study reports an interesting application of the OCT-enabled strain mapping technique using  the vector method for studying osmotic deformations in cartilaginous samples. It is a proof of concept study that shows how spatially-resolved quantitative characterization of osmotically-induced deformations can be measured.

However, in its present form, this reviewer finds that the manuscript is strongly lacking in context, motivation, and precision/details in the discussion. Below are some suggestions to revise the manuscript, related to these issues.

The authors present a concise summary of OCT-based methods, however, no comparisons are made with other existing strain mapping methods. This is found to be a significant weakness in the paper because little context and motivation is provided to evaluate the relative significance of the measurements that can be characterized by OCT.

Overall, the writing should be edited significantly to be more precise. For example, in the discussion, “safety and efficiency” is alluded to without context or explanation. The authors allude to their results showing that highest deformations are observed in the initial stages of diffusion, however the discussion of these results is cursory at best. Can the authors infer something perhaps about the mechanisms driving these results? The discussion section in general reads like a literature review rather than an evaluation of the presented results and their implications. In the first few paragraphs, the discussion reads like a comparative survey: it is hard to distinguish the significance of presented work as a result. In the later paragraphs, the discussion of the results is very hard to follow and should be revised.

Numerous grammatical errors are found throughout the manuscript, and need to be corrected.

A conclusion section is missing in the manuscript.

Author Response

Reviewer 2– General comment

This paper presents an application of phase-resolved Optical Coherence Elastography (OCE) to visualize transient local strains in biological tissues. Applications of this technique are demonstrated for studying the dynamics of osmotically-induced strains in cartilaginous tissue with optical clearing agents. The technique is able to resolve transient spatially-inhomogeneous strain fields excited by the clearing agents, to reveal details of complex spatio-temporal evolution of alternating-sign osmotic strains during agent diffusion. Differences among different clearing agents are discussed with respect to mechanical mechanisms associated with real-time kinetics.

This study reports an interesting application of the OCT-enabled strain mapping technique using  the vector method for studying osmotic deformations in cartilaginous samples. It is a proof of concept study that shows how spatially-resolved quantitative characterization of osmotically-induced deformations can be measured.

However, in its present form, this reviewer finds that the manuscript is strongly lacking in context, motivation, and precision/details in the discussion. Below are some suggestions to revise the manuscript, related to these issues.

Authors: We thank Referee 2 for the effort and time spent for assessing our manuscript and greatly appreciate the constructive feedback and suggestions which helped us improve the quality of the manuscript. In the revised manuscript all introduced changes are highlighted by blue-color font.

Specific comments:

1) Reviewer: The authors present a concise summary of OCT-based methods, however, no comparisons are made with other existing strain mapping methods. This is found to be a significant weakness in the paper because little context and motivation is provided to evaluate the relative significance of the measurements that can be characterized by OCT.

Authors: We are thankful for addressing to this important issue. The motivation has been given in revised manuscript in more detail. We essentially edited the Introduction and supplemented it with more explicit comparison with other elastographic techniques. Also we somewhat modified the structure by only briefly characterizing the used OCE method in the Introduction and replacing its discussion in more detail in Section «Materials and methods».

Introduction is supplemented with the following: “In the last decades elastographic imaging technologies that emerged in 1990s have become widely used in various biomedical applications. Such elastographic modalities are based on some methods enabling structural imaging; presently, these are first of all medical ultrasound (US) [1] and magnetic resonance imaging (MRI) [2]. The main medical application of elastographic techniques is characterization of tissue stiffness (i.e., shear or Young’s moduli), to enable which the basic US or MRI-based imaging is supplemented to some auxiliary mechanical action (either quasistatic or transient) applied to the studied tissues/organs. The transient-type (shear-wave-based) elastography is implemented in several US platforms enabling ultrasonic visualization of the propagation of auxiliary shear waves in the tissue. In these methods visualization of genuine local strains is not required at all. For elastographic techniques based on quasistatic auxiliary loading, an essential stage of elastographic characterization involves mapping of local slowly varying strains in the studied region. This approach is often called “strain elastography”, so that methods of strain visualization are often called “elastography” even if strains are not necessarily produced by mechanical loading. Visualization of strain requires obtaining of a series of structural scans of deformed materials/tissues. The displacements of particles in the so-obtained compared images are then recalculated in genuine strains by finding local gradients of the interframe displacements. Due to inevitable measurement errors at both stages (initial reconstruction of displacements and subsequent finding of strains) the resolution of the resultant strain maps is at least several times (and even an order of magnitude) lower than the initial resolution of structural images. Correspondingly, for the conventional US-based and MRI-based elastographic imaging, the resolution is on the order of several millimeters and even lower.

In this context, new possibilities can be enabled by another visualization technique, Optical Coherence Tomography (OCT) that occupies an intermediate niche between the high-resolution optical microscopy and medical ultrasound from the viewpoint of resolution and sizes of the imaged area. OCT scans represent the distribution of optical-backscattering strength; in the appearance OCT-scans are rather similar to US-scans, but enable much higher, micrometer-scale resolution. In view of this similarity, by analogy with elastographic approaches that were proposed in the beginning of 1990s in medical ultrasound, utilization of OCT for studying microscopic deformations of biological tissues and assessing their biomechanical properties was proposed ~two decades ago in the seminal paper by J. Schmitt [3].”

And further: “In comparison with US-based and MRI-based elastographic methods OCT-based elastographic techniques opened a broad range of previously unavailable possibilities due to the intrinsic much higher resolution of OCE. Since OCT in the recent decades has become a standard imaging technique in ophthalmology [4], it is not surprising that OCT-based elastographic strain imaging has also been tested in various ophthalmic applications [5-10] and presently is actively studied for oncology-related applications as well [11-13]”.

Additionally, the concept of the used OCT modality is regarded in the revised manuscript more precisely in comparison to the other medical elastographic techniques. The following is given in Introduction: “In the present study, we use an advanced realization of the phase-resolved OCT-based strain mapping. The used method is termed “vector”, because it operates with the OCT-signals represented as vectors in the complex-valued plane. In more detail the used phased-resolved realization of mapping local strains and the features/advantages intrinsic to its vector form are explained in the next section “Materials and Methods”. The main point is that the OCT-based mapping of local strains enables much higher spatial resolution than that of conventional medical elastography methods (US-based and MRI-based). Thus OCE opens previously unavailable prospects to quantitatively visualize in real time the osmotic-strain dynamics accompanying penetration of diffusion fronts with a spatial resolution of several tens micrometers with total observation time of hundreds seconds and, if necessarily, hours. The characteristic sizes of the imaged area in OCE are the same as in structural OCT imaging – on the order of several millimeters laterally and ~millimeter in the tissue bulk.

    Since OCT-based strain imaging may be used to visualize not only mechanically produced strains it has been tested to visualize some other types of deformation processes, e.g. tissue drying, heating, etc. [29].”

2) Reviewer: Overall, the writing should be edited significantly to be more precise. For example, in the discussion, “safety and efficiency” is alluded to without context or explanation.

Authors: The important issues of safety and efficiency of OCA application mentioned by The Reviewer and related osmotically-induced mechanical effects in tissues have been regarded in Discussion section in more detail along with introducing new citations. Discussion is supplemented with the following:

“In this context, one should distinguish the average (“global”) deformations of the macroscopic tissue volume subjected to optical clearing and local spatially non-uniform strain variations caused by high gradients of OCA concentration. The first process involves the alteration of the tissue macroscopic parameters, such as thickness and length or bulk moduli [41-45,48]. For instance, cartilaginous tissue demonstrates extensive shrinkage after ~1.5 h application of clearing solutions of glycerol and glucose [41] which is already available for visual assessment and can be directly measured by conventional optical microscopy ruler with sufficient accuracy. The knowledge of such a durable cumulative effect of OCA on the tissue structure is of upmost importance for estimation of its properties on the final stage of clearing which is close to the equilibrium state in terms of diffusion. However, at the first minutes of OCA diffusion the distributions of water and agent concentrations are strongly non-uniform [49,50]. Additionally, the so-called diffusion fronts occur, which may be accompanied by intense straining and dramatically alter tissue mechanics, though the existence of these transient effects is comparatively short-term [47]. These non-equilibrium effects are insufficiently studied up to the date due to complexity of theoretical analysis and the lack of physical methods enabling their direct observation. As shown in present work for cartilage and in [47] for cornea, such osmotically-induced non-uniformities of strains and displacements can emerge on a scale of tens of micrometers. In fact, the effects of the non-equilibrium strain fields on cell viability, matrix integrity and sustainability of different tissues have never been directly observed before. It may reasonably be expected that the duration of a non-equilibrium state is dependent on the size of a particular agent molecule. This expectation is supported by the present study, in particular Fig. 3f demonstrated that the osmotically-induced positive strain fields in case of small glycerol molecules become nearly saturated within 200-350 s intervals, whereas for Omnipaque, strain still continues to gain intensity (Fig.3f). For dextran, a high molecular weight compound studied in [47], it takes tens of hours to proceed through maximum strain intensity caused by dehydration and dextran integration. It was shown previously on isolated chondrocytes and 3D cartilaginous scaffolds that alterations of micromechanical properties of the matrix tissue surrounding cells may influence to a great extent to the cell signaling and overall tissue homeostasis [51-53]. The recently found mechanotransduction pathways in cartilage attract increasing attention during the recent years [54]. Moreover, osmotic stress may influence the metabolic activity of cells [55], in extreme cases leading to cell death [56]. In this regard, the method of direct and non-destructive monitoring of transient mechanical strains demonstrated in present work may greatly contribute to the evaluation of the particular correlations between the osmotically-induced strain duration and intensity and biochemical response of the tissue. Moreover, it is noteworthy that the method is not limited to cartilaginous tissues and the scope of its applications can be significantly expanded to other tissues and organs where superficial scanning is accessible.”

3) Reviewer: The authors allude to their results showing that highest deformations are observed in the initial stages of diffusion; however the discussion of these results is cursory at best. Can the authors infer something perhaps about the mechanisms driving these results?

Authors: Although the present work is in general a pilot study of the method capabilities, we find Reviewer’s interest in possible mechanisms quite reasonable and have provided a more detailed discussion of the strong alternating-sign character of strain at the initial stage of OCA diffusion in Discussion section. Additional latest literature related to the phenomenon is also introduced in the revised version of the manuscript. For the presently available data, the description/discussion of the results are given as detailed as possible. We hope that future research will help us to further elucidate the mechanisms underlying the observed phenomena.

In particular, the Discussion is supplemented with the following: “Concerning possible reasons of this one can argue that, first, the hindering for glycerol permeation into the tissue matrix due to its dehydration-related shrinkage leads to its anomalous accumulation at the subsurface level and subsequent tissue dilatation. The second reason can be connected with the structure of cartilaginous matrix itself. The porous matrix of hyaline-type cartilage consists mainly from collagen and proteoglycan aggregates integrated with each other according to the network principle [60]. Proteoglycan molecules possess the negatively charged sulfate and carbonate groups that are fixed and in normal condition compensated with positive ions of interstitial fluid. The violation of matrix electroneutrality due to, for instance, compressive load and outflow of the fluid with compensating ions from the compressed area, gives rise to a series of forces striving to restore the equilibrium, which include streaming and diffusion currents, changing of the macromolecule conformation and even piezoelectric effects [61]. Considering the phenomena of alternating-sign character of strain in subsurface cartilage layer during the first minutes of glycerol diffusion observed in the present study, one can speculate that the strong subsurface dehydration of cartilage caused by application of glycerol may promote the outflow of positively-signed ions as well that can no longer compensate the negatively-signed cartilage proteoglycan groups. Further, the electrostatic repulsion of the “naked” charges of proteoglycans may cause the process similar to that known as cartilage osmotic regulation [60,62,63]. The electrostatic repulsion of the liberated fixed charged groups may provoke the dramatic structural extension shown in (Fig.2c; Fig.3b,d)”.

4) Reviewer: The discussion section in general reads like a literature review rather than an evaluation of the presented results and their implications. In the first few paragraphs, the discussion reads like a comparative survey: it is hard to distinguish the significance of presented work as a result.

Authors: The Discussion section is now supplemented with more details concerning the issues discussed in previous two answers, in particular, the mechanisms and safety and efficiency concerns. Overall, the section has been substantially revised and we believe the revised version looks more precise and convincing.

5) Reviewer: In the later paragraphs, the discussion of the results is very hard to follow and should be revised.

Authors: The later paragraphs discussing the general underlying mechanisms of sign-alternating deformations have been extensively revised (see also the answer to question (3)), we tried to make them look clearer and more consistent.

6) Reviewer: Numerous grammatical errors are found throughout the manuscript, and need to be corrected.

Authors: The manuscript has been subjected to extensive correction of grammatical errors and misprints.

7) Reviewer: A conclusion section is missing in the manuscript.

Authors: Conclusion briefly summarizing the main results has been added (Sec.5): “The application of the new OCE modality to observe non-uniform osmotically-induced strain fields in biological tissues is presented. OCE enables one to monitor the strain alterations as small as 10-4 and as large as 0.4-0.5 for axial strain value. The duration of uninterrupted observation can be tens of minutes which allows one to cover the whole range of the most intense strain accumulation during OCA diffusion into biological tissue. The concentration-dependent effects studied for glycerol impregnation into cartilaginous tissue include the low intense subsurface dilatation for concentrations less than 30-35% and alternating-sign strain field for highly concentrated glycerol. The effect of Omnipaque was found to be similar to that of glycerol, but proceeding with substantially slower rate.”

Round 2

Reviewer 2 Report

The manuscript has been greatly improved with the added edits.